# Feasibility of a self-management intervention to improve mobility in the community after stroke (SIMS): A mixed-methods pilot study

Ahmad Sahely[1]*, Carron Sintler[2], Andrew Soundy[3], Sheeba Rosewilliam[3]

**1** Physical Therapy Department, Collage of Applied Health Sciences, Jazan University, Jazan, Saudi Arabia, **2** University Hospitals Birmingham NHS Foundation Trust, Queen Elizabeth Hospital Birmingham, Birmingham, United Kingdom, **3** School of Sports, Exercise and Rehabilitation Sciences, University of Birmingham, Birmingham, United Kingdom

* asahely@jazanu.edu.sa

**Data Availability Statement:** All relevant data are within the manuscript and its Supporting Information files.

## Abstract

### Objective

To evaluate the feasibility of implementing a self-management intervention to improve mobility in the community for stroke survivors.

### Methods

A two-phase sequential mixed methods design was used (a pilot randomised controlled trial and focus groups). Participants were adult stroke survivors within six months post discharge from hospital with functional and cognitive capacity for self-management. The intervention included education sessions, goal setting and action planning, group sessions, self-monitoring and follow up. The control group received usual care and both groups enrolled for 3 months in the study. Feasibility outcomes (recruitment and retention rates, randomisation and blinding, adherence to the intervention, collection of outcome measures, and the fidelity and acceptability of the intervention). Participants assessed at baseline, 3 months and 6 months for functional mobility and walking, self-efficacy, goal attainment, cognitive ability, and general health. A descriptive analysis was done for quantitative data and content analysis for the qualitative data. Findings of quantitative and qualitative data were integrated to present the final results of the study.

### Results

Twenty-four participants were recruited and randomised into two groups (12 each). It was feasible to recruit from hospital and community and to deliver the intervention remotely. Randomisation and blinding were successful. Participants were retained (83%) at 3 months and (79.2%) at 6 months assessments. Adherence to the intervention varied due to multiple factors. Focus groups discussed participants' motivations for joining the programme, their perspectives on the intervention (fidelity and acceptability) and methodology, perceived improvements in mobility, facilitators and challenges for self-management, and suggestions for improvement.

**Funding:** The author(s) received no specific funding for this work.

**Competing interests:** The authors have declared that no competing interests exist.

## Conclusion

The self-management intervention seems feasible for implementation for stroke survivors in the community. Participants appreciated the support provided and perceived improvement in their mobility. The study was not powered enough to draw a conclusion about the efficacy of the program and a future full-scale study is warranted.

## Introduction

Every year, stroke affects more than 100 000 individuals in the UK [1]. Following a stroke, about half of the people who survive need help with their daily activities as a result of various impairments that affect their physical, cognitive, emotional, and social well-being [2,3]. A major physical disability following stroke involves limitations in an individual's lower limb activities such as balance and mobility [3,4]. Mobility has been found to be a key problem for two-thirds of people after stroke and can lead to depression and isolation [5,6]. Rehabilitation is deemed to be essential for the recovery process after stroke as it helps to improve the impaired functions resulting in a better quality of life for stroke survivors [7].

Improvements in functional outcomes can be facilitated by increased intensity of therapy especially during the first 3–6 months after stroke as the neurological and functional recovery are optimal in this period [8,9]. However, evaluation of current practice in the UK shows that the intensity of therapy delivered for stroke rehabilitation in the stroke units is less than optimal (11.24 (SD = 7) minutes vs. 45 minutes) and 45% of stroke survivors might not receive adequate therapy after leaving hospital [10–12]. The situation had worsened due to the impact of COVID-19 on health services [13]. During the pandemic, stroke survivors were rushed for early discharge from acute care with limited support in the community, due to services that had stopped completely during lockdowns. Survivors had to rely on their carers, and their individual ability for self-management as the only way of coping with their conditions while lacking formal support [14].

Prior to the pandemic, it has been evident that the delivery of a higher intensity therapy and support for survivors with mild to moderate impairments post-stroke can be provided through self-management interventions [15,16]. Self-management (SM) has been recommended as an approach that empowers the role of patients in facilitating their recovery in addition to the usual care provided by professionals [17]. SM interventions use strategies such as patient education, involvement in goal setting and action planning, peer support, and self-monitoring to enhance individual's independence in managing their therapy plans [18]. In the last decade, various SM interventions have increasingly shown a positive effect on survivors' self-efficacy, emotional and functional recovery, and social participation [15,16,18–21]. However, most of these studies used only a generic outcome such as self-efficacy, quality of life or physical activity and applied SM interventions that involved extensive involvement of the therapists over a large number of therapy sessions and follow up; such use of resources is unrealistic in practice. There still is a need for development of effective interventions that apply self-management principles for rehabilitation of specific outcomes such as mobility.

Overall, SM interventions for improving the mobility of stroke survivors in the community are uncommon [22]. Recently, there have been some studies that used SM, some of which incorporated interventions to improve mobility outcomes after stroke. Examples of these interventions include the Independent Mobility-related Physical Activity (IMPACT) Program [19], the Rehabilitation Training (ReTrain) intervention [23], the Extended Rehabilitation

Service (EXTRAS) for stroke patients [24], and the Bridges self-management programme [20]. The results of the previous studies confirmed that SM interventions delivered for individuals or in groups in the community can be effective for people after stroke. However, most of the interventions used in these studies were criticised for their high resource requirements and ambiguity regarding the theoretical underpinnings, factors that make them challenging to implement in the NHS system. Therefore, development of a comprehensive SM mobility intervention that can be realistically embedded within the current practice of stroke rehabilitation has been suggested [15]. To the best of the authors knowledge no further studies have addressed this gap following the publication of the review. Work was undertaken to address the gap in the current practice by developing a self-management intervention that focused on mobility training in the community (SIMS). The SIMS was developed based on evidence from a systematic review of the literature about self-management for mobility rehabilitation following stroke [15] and in consultation with senior stroke survivors and NHS clinicians as an effort for knowledge translation into practice. The description of the SIMS is provided in the intervention section.

## Aim and objectives

To examine the feasibility of a newly developed self-management intervention for functional mobility for stroke survivors in the community (SIMS).

The specific objectives were:

1. To evaluate the feasibility of implementing the new intervention in stroke rehabilitation services in the local community using a mixed-methods study design.

2. To explore participants' perspectives about the acceptability, practicality, and fidelity of the new intervention.

3. To scope out methodological feasibility for a future randomised control trial.

## Methods

### Design

A two-phase sequential explanatory mixed methods design was undertaken including a feasibility randomised controlled trial (two arms, assessor blinded) followed by focus groups. The first phase included quantitative methods examining the feasibility of the intervention using standard outcome measures while the intervention was carried out to the participants. This then followed by qualitative focus groups with participants to get deeper understanding of their lived experience within the study and to suggest direction for future research. The mixed methods approach was used to elaborate, complement and triangulate findings from each method [25]. The study was conducted between December 2021 and March 2023. The CONSORT 2010 statement: extension to randomised pilot and feasibility trials [26] was used to guide the conduct and reporting of this study.

### Participants and sampling

Participants were purposefully selected using a criterion-based sampling as they needed to have a certain level of mobility and cognitive ability to participate in this study. Participants were eligible if they; (a) were adults (18 years to 99 years), (b) diagnosed with a first stroke, (c) within six months post-discharge from the hospital, (d) with Functional Ambulation Category (FAC) $\geq$ 2 (at least able to walk with a maximum of one person assistance, with or without a

gait aid [27], (e) had cognitive capacity to communicate and consent to participate in the study (screened using MoCA), (f) could speak or had a family member who spoke English, and (g) had access to a suitable electronic device to meet on Zoom if they had to join the study online.

Patients were excluded if they were medically unstable or had other major co-morbidities that might be a risk for community walking, severe cognitive impairments, severe aphasia, severe spasticity or severe arthritis.

## Setting

Participants were recruited from different teams (i.e., early discharge, and community service teams) a rehabilitation hospital in Birmingham, UK and online through Stroke Association and Different Stroke websites. We recruited participants from multiple teams since this has the advantage of ensuring the study protocol and intervention can be widely implemented and tested on participants with diverse levels of clinical trial experience and resources [28]. All the study activities took place in participant' homes, online via Zoom, or in the school of Sport, Exercise and Rehabilitation Sciences at the University of Birmingham. Data collection (intervention and assessments) took place between December 2021 and March 2023.

## Sample size

As no formal sample size calculations are required for feasibility studies and the objectives of trials focused on feasibility can often be met with relatively few numbers of participants, we aimed to recruit 90 participants (45 per group) [29,30].

## Randomisation, approach and consent procedure

Stroke survivors were approached by their therapists at the sub-acute rehabilitation hospitals and through referrals made by Early Supported Discharge (ESD) team or the community therapists. Eligible Participants, who showed interest in taking part in the study to their therapists, were provided with a consent to contact form for further contact by the research team. The researchers contacted the eligible participants to check their readiness in joining the study after providing them with information sheet, data management plan and obtaining consent for their participation. After providing a written consent, the participants were assessed at baseline and then randomly allocated to treatment or control groups by an administrator using computer-generated randomisation chart. No restrictions were applied on randomisation blocks.

## Intervention

The intervention group received the SIMS in addition to any usual care provided to the participants by the NHS. All components of the SIMS were delivered for the intervention group in their homes when they were visited by the research team or through online zoom meetings over a period of 12-weeks. Self-monitoring was done by participants using a pedometer and a recording diary to record their daily training and number of steps. Details of the SIMS protocol are provided below.

### Week 1: (Group and Individual)

• Education about impact of stroke, improving gait, self-management and safety.

　• Self-management strategies including individualised goal setting for mobility goals and action-planning using exercises from a booklet based on their baseline assessment. The

exercise booklet was developed based on evidence and had a selection of appropriate exercises for strengthening, balance and endurance improvement.

• Patient were taught home exercises using the same exercise booklets and self-monitoring of mobility training was done using a recording diary. A safety tips documents with resources for guidance was provided prior to the start of exercise plan.

### Week 2, 4, 6, 8 &10: (Group)

Participants had an online meeting every two weeks for group exercise sessions and peer support was provided by an experienced stroke patient in one of the weeks. The researcher conducted one group session per week for 10–12 people where participants are taught exercises for strengthening, balance, and mobility from the exercise booklets.

To improve safety in zoom classes, a qualified researcher and a registered therapist were both present to deliver the classes.

Patient-experts will provide peer-support during the group sessions. The patient-experts are stroke-survivors who were paid from the PPI costs.

### Week 3, 5, 7, 9 &11: (Individual)

A follow up call was made by the research therapist for reviewing goals, updating exercise plan and patient's progress in the programme every fortnight.

### Week 12: (Group)

At week 12, a reiteration of safety, review of mobility-training and positive reinforcement were discussed with each participant for long term sustainability.

### Control

The participants of this group received an education session about walking safely at the beginning of the study (week1) using a power point presentation. Due to the COVID pandemic the education sessions were delivered through zoom unless patient requested a physical presentation at their homes. They set goals at the start of the study along with the researcher. The control group also received the routinely provided care by the community services for stroke survivors after their discharge from the stroke unit. The usual care in this context is guided by local practice in the NHS and is expected to follow the national clinical guidelines for stroke care.

### Outcomes

Participants demographics and stroke related information such as age, gender, height and weight, stroke type, time since onset, affected side of the body, use of mobility aid, presence of carer and GP addresses were collected at the baseline.

Feasibility outcomes included: recruitment rate (proportion of eligible participants who enrolled in the study) and validity of eligibility criteria (number of participants eligible/not eligible for each criterion), feasibility of randomisation and blinding and ability to reduce contamination, retention rates (proportion of enrolled participants who completed the study), fidelity and acceptability of the intervention (lived experiences and perspectives of participants from focus groups), participants' adherence to the intervention and follow up (the number of days the participants did their exercises and recording diary) and feasibility of the selected outcome measures. Participants from both groups were asked to describe what they had received

from the NHS as usual care for their rehabilitation at baseline and at the end of their participation.

Focus groups used an interview guide that included questions to explore acceptability, practicality, sustainability, effectiveness, and ways to improve intervention from participants' perspectives.

Assessments included outcomes related to the constructs of self-management such as self-efficacy and goal-achievement in addition to mobility related physical and psychosocial outcomes. Self-efficacy was measured by using the stroke self-efficacy questionnaire [31]. The Goal Attainment Scale (GAS) was used to measure participants ability to achieve their goals [32]. The Patient Specific Functional Scale was used to assess participants' ability to perform specific daily activities related to mobility [33]. Level of adherence and performing of other self-management related elements were examined using the Patient Specific Functional Scale, the recording diaries of physical activities, and action planning documents. Adverse events had been reported throughout follow up calls (biweekly) and recording diaries of daily training.

To determine the impact of SIMS on participants' mobility; the Functional Gait Assessment (FGA), Timed Up and Go, 10-meter Walking Test, Six Minutes Walking Test, number of steps walked per day using pedometer were used. These outcome measures have been tested for reliability and validity to detect changes in stroke population [34,35]. To examine the cognitive and psychological well-being, the Montreal Cognitive Assessment (MoCA) and General Health Questionnaire (GHQ-12) were used [36,37].

The selected outcomes were measured at study baseline and three months later (at the end of the intervention) followed by a follow-up assessment at six months from the start of the study. The assessments were collected by an independent blinded assessor and only the first author (AS) and his academic supervisor (SR) had access to information that could identify individual participants during or after data collection.

The participants self-selected themselves to participate in focus groups after study completion.

## Ethics and registration

Ethics approval for this study was initially obtained in June 2020 from the Health Research Authority (HRA) at the NHS (IRAS project ID: 269079). However, due to the situation of the COVID-19 pandemic, the research team applied for an amendment to the approved protocol to allow for online delivery of the intervention and to recruit participants directly from the community and a final approval was received in June 2021. The study was registered prospectively on the ISRCTN registry (ISRCTN15728885) in September 2021.

## Data analysis

A descriptive statistical analysis was used to describe the demographic data, results of the feasibility and clinical outcomes [38,39]. Categorical data was summarized using frequencies and percentages. Continuous data was summarized using means, standard deviations or ranges.

The aim of the data analysis for the efficacy outcome measures was to evaluate the feasibility of using these outcome measures, keeping in mind that feasibility trials are underpowered to detect clinically significant treatment effects [28]. Hence, the p and eta square values were not provided for the change in outcome measure following the guidance of methodological literature [28,40]. Microsoft Excel Version 2016 was used to carry out the descriptive analysis of quantitative data. The feasibility of conducting this study was determined by some pre-defined criteria following the RAG (Red, Amber, Green) rating system described by Avery et al., 2017 [41]. This includes an acceptable rates of recruitment, retention and participants adherence to the SIMS intervention.

### Recruitment rate: participant response

- Red: if < 25% of eligible patients were recruited.

- Amber: if 25–50% of eligible patients were recruited. If recruitment rate is low, then it is possible to approach patients referred to the early supported discharge team within the trust and apply an online recruitment.

- Green if > 50% of eligible patients were recruited.

Retention rates (at 3 and 6 months)

- Red: Follow-up rate < 50%

- Amber: Follow-up rate 50–70%. In case of high attrition we will explore reasons and modify barriers

- Green: Follow-up rate > 70%

  Adherence to the intervention

- Red: if < 30% patients.

- Amber: if 30–50% patients. As a plan, adherence will be encouraged through periodical telephone reminders and group meeting.

- Green: if > 50% patients.

For the qualitative data from focus group interviews, a qualitative content analysis approach was used to present the identified themes from participants' discussion about the SM intervention [42,43]. The data from focus groups were coded by the first author (AS) and themes then built in an iterative process. The other two authors (SR and AS) have independently reviewed the codes and themes and discussed their perspectives in a meeting with first author.

The findings from focus groups were integrated with results of the quantitative outcomes in the results section by matching themes with the relevant quantitative data. The integration of qualitative and quantitative data has been suggested in mixed methods studies to provide a comprehensive presentation of findings from both approaches addressing the same objectives rather than presenting them separately [44].

## Results

The results of feasibility outcomes; recruitment and validity of eligibility criteria, randomisation and blinding, retention, fidelity and acceptability of the intervention, and feasibility of selected outcomes are presented with relevant themes/sub-themes derived from the qualitative data in this section. There were six themes derived from focus groups. Themes included 1) motivations for joining the programme, 2) participants' evaluation of the programme, 3) perceived benefits, 4) facilitators of self-management, 5) challenges for participation in self-management, and 6) suggestions for improvement. Participants' quotes are provided anonymously to support the results of qualitative findings as needed. The integration of quantitative and qualitative data relevant to each feasibility objective is summarised in Table 1.

### I) Feasibility of recruitment and validity of eligibility criteria

**Participant enrolment.**   Screening and recruitment were carried out between November 2021 and August 2022. A total of 451 patients were screened by four care teams at the hospital setting. Thirty-eight patients (8.42%) of the 451 screened were eligible. Of these 38 eligible

**Table 1. Integration of findings in relation to feasibility objectives.**

| Feasibility objectives | Quantitative data | Themes and sub-themes |
|---|---|---|
| I) Recruitment rate and validity of eligibility criteria | • Participant enrolment<br>• Validity of eligibility criteria (participant exclusion)<br>• Participants characteristics | Theme1) Motivations for joining the programme with the subthemes<br>• the hope for improvement and independence,<br>• helping themselves and others by taking part in research,<br>• not receiving enough care from the NHS and<br>• self-management is a part of journey post-stroke |
| II) Randomisation and blinding | Number of participants randomised<br>Ability to blind assessors and reduce contamination | Theme 2) Participants' evaluation of the programme with the subtheme perspectives on the randomised design |
| III) Retention rates | • Number of enrolled participants who completed the study<br>• Adverse events | Theme 2) Participants' evaluation of the programme with subtheme<br>• factors influencing continuation in the programme |
| IV) Fidelity and acceptability of the intervention | • Delivery of the components of the intervention i.e. education sessions, goal setting and action planning, online group meetings, and follow ups<br>• Reporting of usual care and how the intervention fits with usual care | Theme 2) Participants' evaluation of the programme with sub themes<br>• Effective protocol elements<br>• Adjunct to usual NHS care<br>Theme 3) Perceived benefits with sub themes<br>• achievement of goals,<br>• motivation to work harder,<br>• improvements in walking,<br>• mental health and<br>• socialisation)<br>Theme 4) Facilitators for self-management intervention related with sub themes<br>• education,<br>• peer support,<br>• technology<br>Theme 5) Challenges for self-management with sub themes<br>• stroke and other health related challenges<br>Theme 3) Perceived benefits with sub themes<br>• Adjunct<br>• Flexibility of programme |
| V) Participants' adherence to the intervention and follow up | • Number of participants who<br>• completed recording dairy,<br>• used pedometer,<br>• Attendance at sessions<br>• Response to the follow up calls | Theme 4) Facilitators for self-management with sub themes<br>• Time for adjustment,<br>• previous knowledge and experience,<br>• social support<br>Theme 5) Challenges for self- management with sub themes<br>• personal factors<br>• technical factors and<br>• environmental factors influencing adherence |
| VI) Feasibility of collecting outcome measures | • Completion of assessments at 3 assessment points. | Theme 2) Participants' evaluation of the programme with sub theme<br>• perspectives on outcome measures |
| VII) Suggestions for future improvement of the programme | _ | • Theme 6) suggestions for improvement |

patients, 17 (44.73%) did not join the study when they were approached (2 admitted to the hospital, 2 went back to work, 3 were found ineligible, 2 had private physiotherapists, 8 refused or thought they were not ready to participate when they were approached). Of the 3 ineligible persons, one person deteriorated and had a poor FAC ($< 2$), the second person had a

cardiopulmonary disease, and the third person had a very poor vision (needing guidance for walking). Twenty-one patients referred by NHS therapists participated in the study. In addition, there was one participant who was referred to us by a previous participant and two participants who joined the study through the Different Strokes website. There were 6 patients from Different Strokes who were excluded after they were found ineligible because they were more than 6 months post discharge from hospital.

Therefore, the total number of participants enrolled in the study was twenty-four. Applying the RAG system, the recruitment was rated as "Green" as more than 50% of eligible participants who were recruited and referred to the research team from the NHS were enrolled in the study. However, the recruitment rate was very low at the study site at screening level which was influenced by COVIID-19 and discussed further in the discussion section. The flow of participants in the study is summarised with reasons for not meeting the eligibility criteria in Fig 1 based on the CONSORT guidelines.

**Participant exclusion.** At the screening stage, numbers of patients excluded for each criterion are described. The highest number of exclusions (34%) was because of not meeting

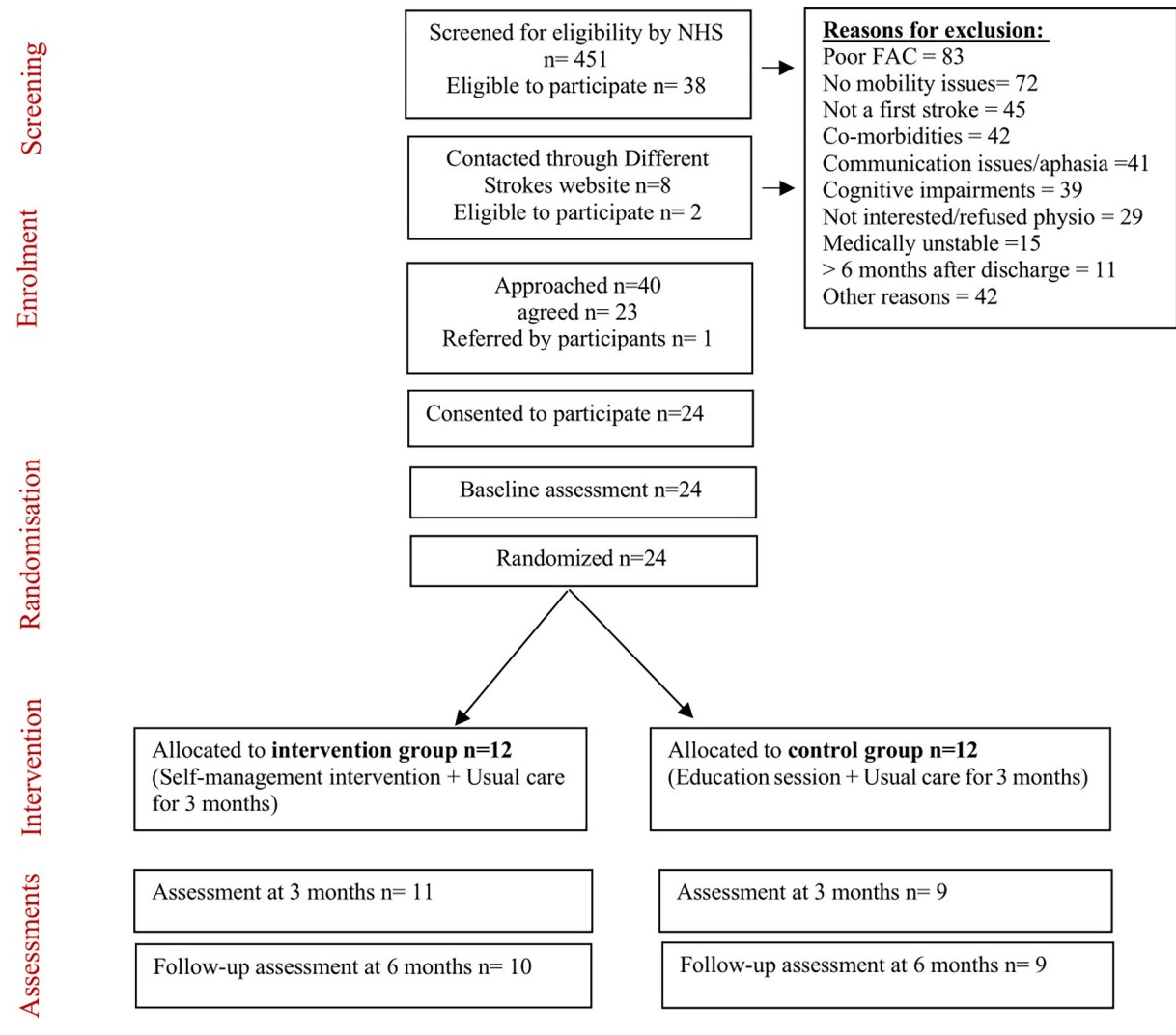

**Fig 1. Flow of participants in the study.**

eligibility for mobility which needed to be at moderate levels to carry out self-management (83 patients had poor mobility and 72 patients had very good mobility scores). Numbers of patients excluded for other reasons were: not a first stroke (45), co-morbidities (42), communication issues/aphasia (41), cognitive impairments (39), medically unstable (15), > 6 months after discharge (11) and other reasons (42) including language, social issues, leaving UK, moved to another care team, back to work, hearing and vision problems. In addition, there were 29 patients (6.4%) who were eligible, but were not interested in the study or refused physiotherapy.

Eligible participants who joined the study revealed certain reasons that motivated them for participating in the study. These motivations are presented with quotes from focus groups below.

**Motivations for joining the programme from theme 1.**   Participants' key motivations for joining the study included a hope for improvement and independence in life post-stroke, and a desire to help themselves and others by taking part in research that can improve their status.

*"I think I'm always happy to help if I can, doing on research—anything—to improve things" (P2, IG).*

*"hopefully, we can pass information over that's benefitted us and hopefully benefits other people." (P5, IG).*

*"one of the problems I've had for a while because of the strokes, has been the balance, and because you were doing a research-type programme on balance. I considered that you would be looking at things slightly different from the normal people, which may mean that you find something that they haven't, that helps me" (P6, IG).*

They were not receiving enough care from the NHS and that lead to their belief that the ability to self-manage their condition is a part of journey post-stroke.

*"I wasn't really getting very much therapy from hospital, so anything that could help me recover was a good shout." (P4, IG).*

*"It fitted in quite well, and the only thing I found was if we were trying to get through to the NHS, you wouldn't be able to get through them that quick." (P9, CG).*

*"Yes, because it just is part of your journey and you improve." (P1, IG).*

**Participants characteristics.**   The sample represented a wide range of characteristics of stroke population in the post-acute stage. Participants aged between 36 and 87 years (mean = 67 (SD = 15.15) years). The number of male participants was higher than females (14 (58%) vs. 10 participants (42%)). Nineteen participants (79%) had an ischemic stroke and 5 (20.83%) had a haemorrhagic stroke. The left side of the body was affected in 14 (58%) participants and the right side in 10 participants (42%). Participants were within 6 months of discharge from hospital (mean time = 4.3 (SD = 1.57) months) except one person who started the programme 7 months after discharge. There were 16 participants (67%) who lived with a carer/family member. The mean scores of outcome measures for both groups at baseline are presented with demographic data in Table 2.

## II) Feasibility of randomisation and blinding

After baseline assessment, all the 24 participants who consented to participate were successfully randomised to one of the two study groups (12 participants each). At post intervention assessment, participants were asked if they knew any other participant in the study and only 2

**Table 2. Participants' demographics and baseline data.**

| | Total N = 24 | Intervention group (n = 12) | Control group (n = 12) |
|---|---|---|---|
| Mean age, mean (SD) | 67 years (15.15) | 63 years (16.53) | 71 years (13.04) |
| Gender (F/M), n (%) | 10 (42%) /14 (58%) | 6 (50%) / 6 (50%) | 4 (33%) / 8 (67%) |
| Height (cm), mean (SD) | 166.35 (9.39) | 166.87 (10.37) | 165.84 (8.73) |
| Weight (kg), mean (SD) | 72.20 (17.89) | 76.59 (18.52) | 67.81 (16.86) |
| Time since stroke (months), mean (SD) | 4.3 (1.57) | 4 (1.83) | 4.5 (1.31) |
| Stroke type (Ischemic/Haemorrhagic), n (%) | 19 (79%) / 5 (21%) | 9 (75%) / 3 (25%) | 10 (83%) / 2 (17%) |
| Have a carer, n (%) | 16 (67%) | 9 (75%) | 7 (58%) |
| Affected side of the body, n (%) | 14 (58%) Left and 10 (42%) Right | 7 (58%) Left and 5 (42%) Right | 7 (58%) Left and 5 (42%) Right |
| Stroke self-efficacy questionnaire, mean (SD) | 86.16 (18.30) | 85.83 (17.97) | 86.5 (19.43) |
| Functional gait assessment, mean (SD) | 17.58 (5.77) | 18 (5.70) | 17.16 (6.05) |
| 10 Meter walking test (m/sec), mean (SD) | Comfortable walking = 0.51 (0.32) | Comfortable walking = 0.53 (0.34) | Comfortable walking = 0.49 (0.31) |
| | Fast walking = 0.60 (0.30) | Fast walking = 0.607 (0.29) | Fast walking = 0.59 (0.33) |
| Timed up and go (sec), mean (SD) | 26.28 (20.14) | 23.15 (12.97) | 29.42 (25.66) |
| 6 minutes walking test (meter), mean (SD) | 141.1 (77.03) | 151.87 (67.66) | 130.29 (87.03) |
| Patient specific functional scale, mean (SD) | 3.75 (1.98) | 3.74 (2.23) | 3.76 (1.79) |
| Montreal Cognitive Assessment, mean (SD) | 22.7 (4.24) | 22.41 (5.51) | 22.91 (2.67) |
| General Health Quesionnaire-12, mean (SD) | 18.04 (6.93) | 16.25 (6.31) | (7.32) |

participants (8.33%) agreed. However, they were both in the same group and were friends before joining the study.

Blinding of the assessors was considered successful as no participant revealed their allocation during assessments and the first assessor was not able to identify any group assignment of participants. Each of the second and third assessors correctly identified 16.66% of assignments when they were asked after the study.

**Participants perspectives on randomised design from theme 2.** Some participants in the focus groups were generally satisfied with the structure of the study which involved a control group receiving minimal help.

*"I thought the structure was good." (P1, IG).*

Two participants from intervention group expressed that they would have had concerns about not receiving the intervention, if they had been assigned to the control group. They suggested comparing two effective treatments instead of no treatment in one group.

*"if you've got two groups surely they should have the same." (P5, IG).*

*"Can I say I don't believe or don't like placebo effects, . . . . . . .. So, you're not really sure whether you're being treated properly or whether you're just being left to let it happen as it happens." (P6, IG).*

*"I think you can have two groups who have different treatments, but they are treatments, and you work out which one is the better one, but not there's one who's not having any treatment at all" (P5, IG).*

## III) Retention

Of the 24 participants enrolled at baseline, 20 (83%) participants (11 from intervention group and 9 from control group) were retained at three months assessment, and 19 (79%) at follow up (10 from intervention group and 9 from control group). Following baseline assessment, 1 participant from the intervention group withdrew as he moved to another country and another participant from the control group died before the 3 months' assessment. Two participants (17%) from control group were not available for post intervention and follow up assessments due to social issues and a fracture of leg (unrelated to participation). At follow up assessment, another participant in the intervention group was lost due to admission to the hospital for unrelated condition (chest infection). The retention rate was rated as "Green" (>70%) of participants at each point of assessment.

**Adverse events.**   There were some reported events that were related to participation each of which are commonly experienced to different levels by stroke survivors. These events included muscle pain in 72% of participants, fatigue (in 53%) and depression because of not achieving some goals (in 12%). Other events (not-related to participation) included a death of one participant the control group after baseline assessment, 2 fractures (1 neck and 1 leg), 5 COVID-19 infections, 1 pneumonia and 1 chest infection.

**Factors influencing continuation within programme from theme 2.**   Participants attributed their motivation to continue in the study to certain elements of the study. For instance, some participants attributed the continuation in the study to the increased understanding of their condition and the programme that was provided throughout the education sessions.

*"I have a better understanding. So, you don't actually give up because you've got a purpose and a goal to aim at, and that's because of the support that you've been given from this understanding" (P6, IG).*

Also, participants liked the level of independence the programme gave them in carrying out their daily training with support of therapist provided if needed.

*"I quite liked the fact that you weren't in my face all the while. Even though I knew you were there, that you weren't hassling, you weren't going how's it going? You set the target. Yes, and you just left us to it." (P10, CG).*

Another element that motivated them to continue was the goal setting introduced to them through the education session provided at the beginning of the programme.

*"It was really easy because it's mostly like target setting, when he was talking about it[education], in your head you could—in my head, I could see what I could possibly do or not do" (P10, CG).*

## IV) Fidelity and acceptability of the intervention

**Delivery of the intervention components.**   Fidelity in this context is described as the extent to which the intervention is delivered as planned in the protocol [45]. This section provides an evaluation of the ability to deliver the components of the intervention (education sessions, goal setting and action planning, group sessions, and follow ups) and qualitative data around, factors that affected the fidelity of intervention (facilitators and challenges for self-management). Acceptability is illustrated by factors relevant to perceived benefits that lead to acceptance of the protocol, presented from focus groups responses.

Over the period of the study (15 months), we were able to deliver 20 education sessions. Most of the sessions were delivered individually for participants in both groups except for 4 participants (17%) who had their education sessions with another participant. Most of sessions were delivered online via Zoom meetings except for 2 participants (18%) from control group who asked for physical delivery of the sessions at their homes. Each session took 25–53 minutes.

Goal setting and action planning were delivered during a home visit for each participant except for 2 participants (9%) who joined the study online from distant geographical area. Each session took around 60–90 minutes. All participants were all provided with hard copies of the exercise booklet, safety tips brochures, recording diaries and pedometers in person for everyone except for those who lived far away in which case these documents were sent by email or post. They also successfully received training on how to use these materials.

For the online group meetings, we were able to deliver 28 sessions (93%) out of 30 scheduled during 15 months of intervention delivery (session/2weeks). Each session took 57–83 minutes. Two sessions (7%) were cancelled because of bank holidays. All the sessions were carried out online although the participants were given the option for physical attendance of sessions. To follow up with participants regarding the goals and exercises plans, 5 calls were scheduled for each participant in the intervention group during their participation (total = 60 calls). We were able to successfully make 82% of these calls (49 calls, 13–27 minutes each).

**Participants' evaluation of the programme from theme 2.**   Participants in the focus groups considered the protocol to be structured and components of the self-management as helpful for patients with complex situations.

> *"I think it was very good, in a way, you have a set programme to go by and trying to work as closely as you can" (P2, IG).*

> *"It has helped me quite a bit really, because I'd had more problems and I found you very helpful." (P8, CG).*

Particularly, participants found the information provided in the education sessions simple to understand.

> *"it was easy to understand, yes." (P8, CG). "the information was obviously good and . . . . . . I get on with them and it's very good the way I think your programme is run." (P5, IG). "it was really easy . . . . . . . . . when he was talking—in my head, I could see what I could possibly do or not do. So yes, it was very good." (P10, CG).*

Some participants positively commented on the methods of communication used by the research team considering participants' individual preferences.

> *"It's good communication, emails, Zoom and text message and actually call" (P1, IG).*

They were positive about the exercises provided within the booklet and the recording diary and desired to continue using them after the programme finished, further indicating acceptability of the study elements.

> *"I managed to get through them all, which I was very pleased with because it's that extra push again, isn't it, to try and motivate yourself into doing a bit more." (P3, IG).*

> *"Also, something that I found very, very useful was filling in the diary. Having that each day to complete, I found that really useful as well as the Zoom sessions." (P3, IG).*

*"So until it becomes the norm I'll continue, and I want to do a lot of things." (P7, IG).*

*"I think in fairness, you were very, very considerate, because there's a lot of the times that I couldn't make and you went out of your way to try and compensate for me, and I appreciated that quite a lot." (P6, IG).*

**Perceived benefits from theme 3.** *A good push for more training.* Participants indicated that they were encouraged by the components of the intervention to increase their training. They considered this 'a *good push for more training '.*

The education session and goal setting were seen to work together to support the planning of therapy.

*"Well, the main thing about that was the understanding of what you were offering and what was wrong with me, because if I have an understanding of what's wrong and what you're offering, I can coordinate the two and it makes more sense then." (P6, IG).*

The programme gave them a plan for the day and enabled self-monitoring which encouraged them to do more training.

*"I think the design of the study is, well, very well organised and it's right in what you've got to do everyday" (P1, IG).*

*"It's given me another a couple of batteries in and I'm setting myself goals again, whereas before I was just going down and down and down" (P7, IG).*

*"I did find that it gave me that extra motivation to exceed my previous scores—which is good —and then week by week I can see, track my improvement. Yes. . ." (P4, IG). "It encouraged you to do a bit more" (P9, CG).*

*"Every morning it introduced me to a pattern where I knew I'd got to do something, and at night recording it. So that, even though it's not a lot, it helped me out." (P10, CG).*

They indicated that they gained more confidence to do their exercises and desired to follow the same protocol after finishing the programme.

*"Sometimes I think you can give up on exercises, and if somebody comes back and helps you to get back onto it you've got that encouragement, because it's hard." (P5, IG).*

They also expressed their fear of losing this motivation after finishing their participation.

*"since we've taken that off, we haven't got that little motivation now" (P9, CG).*

*Goals were reached.* Participants showed their happiness due to achieving goals. Some patients explained that they were encouraged by goal setting to achieve better outcomes such as independence and desired to continue setting goals in their future.

*"Yes, I definitely am. Once I've reached those goals, I want these goals until I get back to as normal as you can." (P1, IG). "I fortunately have managed to. . . As I had my assessment the other day, I was able to. . . I have, yes, achieved pretty well the goals that I wanted to" (P3, IG). "over the time period of the programme, you have an independence to go even to the toilet*

*or just walk around with this. I think I wouldn't have done it, but in the end with this programme we are achieving, and we are moving forward." (P4, IG).* "Oh yes, I did reach those. I think I've told you, I go dancing." (P8, CG).*

A participant indicated that prioritizing of goals at the beginning of the programme might be a challenge and therapist's support can help in overcoming this challenge.

*"At that time, you didn't know what your first priority was." (P9, CG).*

The scores of goal attainment scale (GAS) for the intervention group participants showed that 2 out of 11 participants (18%) achieved their mobility related goals much better than expected ($> 60$) 3 month (end of intervention), 3 (27%) achieved their goals better than expected (50–59), 5 (45%) less well than expected (40–49) and 1 (9%) much less than expected ($< 40$). For control group, there were 2 participants (22%) who achieved their goals much better than expected ($> 60$) at 3 months, 2 (22%) better than expected (50–59), and 5 (56%) less well than expected (40–49). Fig 2 shows the GAS scores for participants in the two groups.

*Walking improvement.* Improvement in walking variables were seen in the result of assessments. Table 3 show the difference in the walking outcome measures (walking speed, distance, endurance) at the second and third assessments. The mean number of steps per day have been increased in the intervention group from 1674.14 steps to 3293.42 steps within 12 weeks. In the control group, the mean number of steps per day has increased from 1790.55 to 3002.77 steps. Fig 3 shows steps per day over time for both groups. Number of participants using assistive devices has been reduced in both groups from 5 to 3 in the intervention group and from 7 to 3 in the control group post intervention. This improvement was sustained till the time of follow up assessment (6 months) in the two groups.

Participants reported perceived improvements in their walking in the focus groups from their lived experience.

*"I've seen the difference, the strength in my leg; it's been brilliant, to be fair" (P1, IG).*

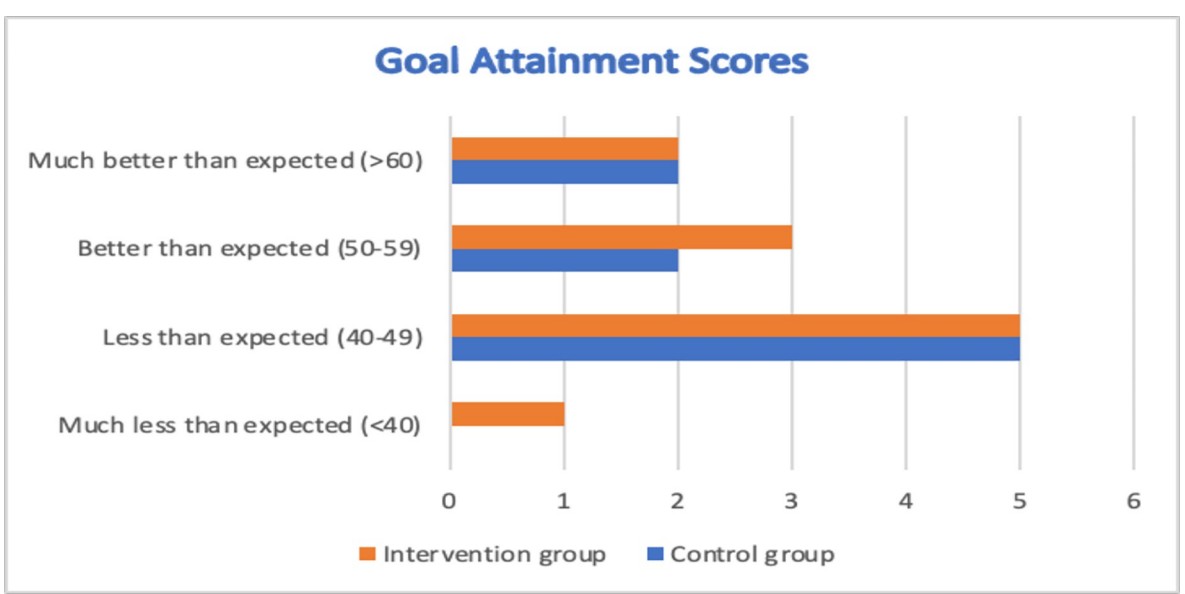

**Fig 2. GAS scores for both groups.**

**Table 3. Mean scores and standard deviations of functional outcome measures at baseline, 3 month and 6 months.**

| | Baseline | | 3 months | | 6 months | |
|---|---|---|---|---|---|---|
| Outcome measuresMean (SD) | Intervention | Control | Intervention | Control | Intervention | Control |
| Stroke self-efficacy questionnaire | 85.83 (17.97) | 86.5 (19.43) | 98.2 (17.22) | 97.44 (18.11) | 93.9 (21.57) | 99.88 (17.81) |
| Functional gait assessment | 18 (5.70) | 17.16 (6.05) | 20.4 (5.01) | 19.33 (6.22) | 20.50 (5.70) | 19.44 (7.17) |
| 10 Meter walking test (comfortable) | 0.53 (0.34) | 0.49 (0.31) | 0.54 (0.27) | 0.54 (0.28) | 0.53 (0.28) | 0.60 (0.23) |
| 10 Meter walking test (fast) | 0.60 (0.29) | 0.59 (0.33) | 0.71 (0.38) | 0.66 (0.38) | 0.70 (0.40) | 0.72 (0.30) |
| Timed up and go | 23.15 (12.97) | 29.42 (25.66) | 20.24 (12.75) | 16.73 (9.09) | 17.94 (10.18) | 16.54 (8.06) |
| 6 minutes walking test | 151.87 (67.66) | 130.29 (87.03) | 153.4 (70.40) | 171.27 (68.49) | 168.7 (67.45) | 179 (75.01) |
| Patient specific functional scale | 3.74 (2.23) | 3.76 (1.8) | 5.16 (2.77) | 6.99 (1.80) | 5.43 (2.39) | 7.14 (1.83) |
| Montreal Cognitive Assessment | 22.41 (5.51) | 22.91 (2.67) | 21.8 (5.31) | 22.77 (3.59) | 22.5 (4.79) | 22.88 (3.62) |
| General Health Quesionnaire-12 | 16.25 (6.31) | 19.83 (7.32) | 13.3 (6.40) | 13.55 (6.48) | 14.10 (6.80) | 14.87 (9.46) |
| Goal attainment scale | - | - | 50.43 (11.25) | 51.81 (12.85) | - | - |
| Use of assistive device | 5 | 7 | 3 | 3 | 3 | 3 |

Particularly, participants mentioned walking more steps without a stick, increased confidence in walking, improved walking beyond NHS care, and that they have Improved because of the programme.

"*I found that very good. I pass my goals kind of thing. I've done 10,000 steps and I feel*

*I could do that easy now daily, but I'm now without my stick now, which is brilliant.*" (P1, IG).

"*I found all of it very useful because I started off just about being able to move with a frame and to walk, and now I can walk in the house just using a stick—which is brilliant really— within just a few months.*" (P3, IG).

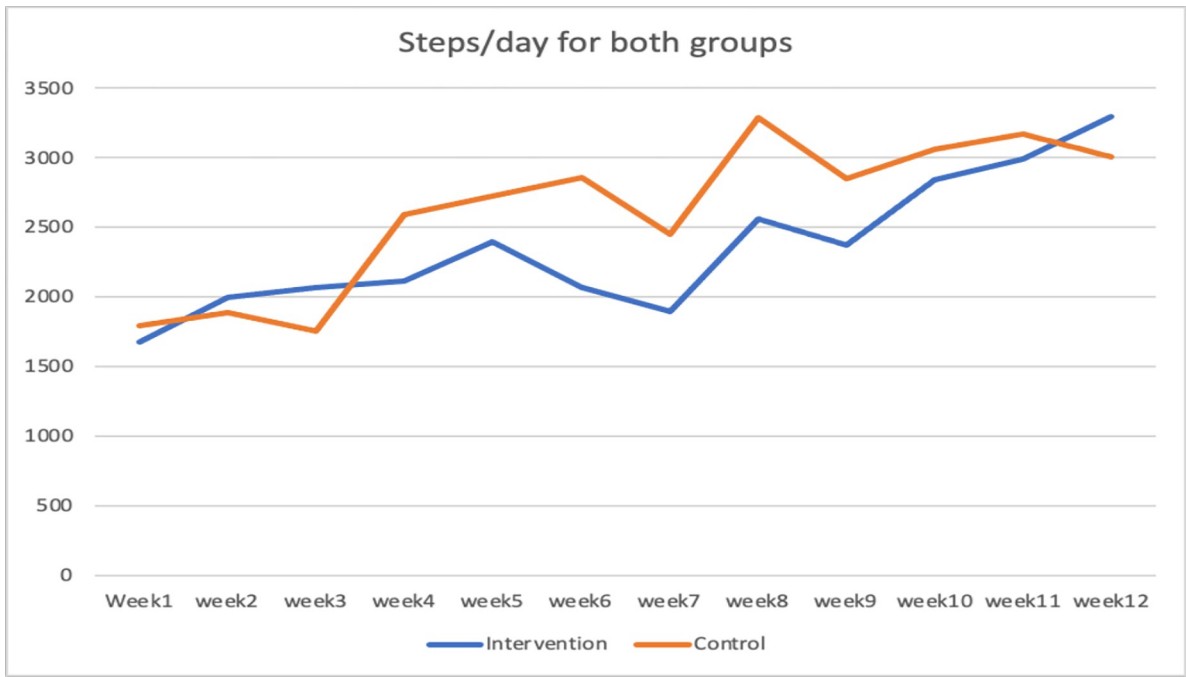

**Fig 3. Steps/Day for intervention and control groups.**

*"Following on from the physiotherapy that I had via the hospital meant that I was able to walk better." (P3, IG).*

*"It does give you a little bit more confidence. I've started to walk again around the road." (P5, IG).*

An improvement in socialisation and participation was also attributed to the programme in the focus groups.

*"It's only because of the programme, I think, we're doing this: it's only because of the programme, you pushing us." (P2, IG). "So yes, and I'm going out now, walking, and*

*I'm socialising more. So yes, I think I have reached the goals that I wanted to." (P8, CG).*

*Benefits of group sessions.* Participants appreciated their participation in the group sessions as they had a chance to meet other people with stroke and share their stories.

*"It was good seeing others, even if they were struggling." (P1, IG). "There's a lot of nice people on there and you could feel the warmth as such and, yes, it was I find everything I've done has been helpful" (P5, IG).*

A participant had shared his study experience with another person who had Parkinsonism and reported that they benefited from the shared knowledge.

*"I've told people about it and I've actually shown them some of the programmes so they could try them out. Some of them . . . . . . . .found some of the sit-down ones very, very useful." (P6, IG). "He was a very advanced Parkinson's disease man. . . . . . . and I was able to show him some of the exercises and some of the things that helped him become. . . Or feel more normal." (P6, IG).*

Participants indicated that they were inspired by talks of senior patients delivered during group sessions.

*"It was really helpful, yes. It was really inspiring as well. More of that, please [laughs]!" (P4, IG).*

Good for mental health

*"it was good for the mental health as well, like you didn't feel you're on your own kind of thing then. That's what I thought was very good." (P1, IG).*

**Facilitators for participation in self-management from theme 4.** Participants identified some factors such as education, peer support and technology that help their participation in this intervention.

They appreciated the value of education sessions provided at the beginning of the programme in helping their understanding of how to manage issues appropriately. This enabled continuation in the programme.

*"Having somebody, a third eye watching you and saying, 'Look, it's not quite right. Can we do it this way'—which is a part of the programme. We can learn from each other." (P2, IG).*

*"Since I have a better understanding, so you don't actually give up because you've got a purpose and a goal to aim at, and that's because of the support that you've been given from this understanding." (P6, IG). "I'll just keep carrying on and doing the exercises, because I think exercise is really important to get your limbs going and to get focused on things." (P8, CG).*

Some participants thought the expert peer support provided within the programme has enhanced their ability for self-management.

*"I thought she was really, really good. You felt encouraged by her and I really do think that those sorts of people, they give you confidence." (P5, IG).*

Participants mentioned some devices that can help self-management of rehabilitation post stroke such as pedometer or physical activity tracker to challenge ability of walking more steps or increasing training intensity.

*"Let's say I had three, 4,000 whatever steps, the next day I wanted to get more, and I think that was a challenge in itself. I haven't got that now, but on my phone I can do the steps and the walking." (P5, IG). "I do feel like I have gained more strength in doing the little exercises, while I was wearing that, because I'd set myself a target." (P9, CG).*

**Challenges for self-management from theme 5.** Participants identified challenges to their participation in the programme some of which were related stroke and other because of other health conditions such as COVID-19 and having other co-morbidities.

Challenges such as speed of recovery, poor level of walking at the beginning, and other stroke related problems (i.e. pain, fatigue, weakness, mood) were commonly reported by the participants as challenges that affected their participation in the intervention.

*"It was a challenge. Some of the exercises I still can't do because, I don't know, the brain to the muscle, that bit." (P1, IG). "I do at the moment still suffer quite badly with fatigue, which I've found a difficult part of the stroke. I've heard about strokes over the years but I didn't realise quite what—how the fatigue can affect you" (P3, IG). "Well, the big one for me was the walking because in the beginning it wasn't very good" (P6, IG).*

Participants who got COVID during or prior to the study demonstrated a huge impact on their recovery and ability for self-management.

*"I was doing six, 7,000 steps a day and I thought, I'm going to crack this not a problem. Then all of a sudden COVID and I wasn't walking as good……… you do lose your confidence." (P5, IG). "I had COVID, I couldn't do as much walking as I wanted to do, but I focused on doing what I could." (P8, CG).*

A participant with a chronic hearing problem had difficulties with balance and only low-level exercises provided to him to ensure safety.

*"When I was 19, they botched an operation in my ear and took all my ear out, and I lost balance completely and it took me three years to learn to walk. I did overcome all that, but since the stroke, I think the balance mechanism in the ear that was damaged has got worse." (P6, IG).*

**Reporting of usual care and fitting in with usual care.** This section shows the feasibility of reporting usual care and how the programme fits with what had been received by participants as a usual care from community services. Reporting of usual care received by participants was obtained from 23 participants. 16 of the participants (70%) received NHS therapy in the community immediately (1–2 weeks) after they discharged from hospital and 7 participants (30%) received therapy after 4–7 weeks of discharge. Only 3 participants (13%) had their NHS therapy concurrently delivered during the study programme.

Most of participants (69.5%) received usual care for 6–8 weeks in the community. Other participants received total of 8–16 weeks of usual care. There were 3 participants (13%) who had a private therapy as a part of their usual care.

Usual care plans included physical and occupational therapy with various of exercise intensity and number of home visits. Home exercises included: standing, walking, leg strengthening, balance, endurance, stairs climbing and descending and upper limb exercises. In addition, some participants reported sensory re-education, teaching the use of assistive devices, and strategies for community reintegration such as walking to local shop or park and back. Most of participants (78%) had received the usual care programmes in their homes. Other participants had their care at home in addition to therapy at rehabilitation centres for some of the sessions where they need some equipment such as treadmill.

*Perceived benefits from theme 3.* The qualitative data showed that the study intervention was considered adjunct to the usual care. In a focus group, participants indicated that the intervention provided an additional help to what received from the NHS. Moreover, the flexibility of being able to fit the study intervention around other NHS care enabled them to manage their schedules alongside NHS appointments.

> *"I thought it was excellent additional help" (P10, CG).*

> *"it was good because I could let you know what I was doing because I knew the week before my appointments and things like that. So I was always okay and the private physio was always on a Saturday. NHS was usually a Monday, Tuesday or Wednesday and if you were a Wednesday I'd be in touch with them, 'Can I have that Tuesday' or something. So it's been flexible" (P1, IG).*

> *"It actually fits in quite well. I'm just pleased that I'm doing it really, because it gives you so much support." (P8, CG).*

Other participants who finished the NHS therapy before joining the study considered the programme as a continuation of what was received from the NHS.

> *"I had three different physiotherapists visit me, which was very good. So, therefore the programme has followed on from that so it's been just a continuation for me—which has been brilliant really, thank you, yes." (P3, IG).*

## V) Participants' adherence to the intervention and follow up

**Adherence to the recording diary and use of pedometer.** Recording diary was fully completed by 10 (43%) of the participants from both groups (70% from control group), 2 provided uncompleted diaries (10–60% missing days), 4 provided only the number of steps/days through email. There were 2 participants who completed but said they lost them before handing them to the researchers and other 5 participants (22%) did not return their recording diaries.

Nine participants (39%) used the pedometers provided by the research team, 1 participant (4%) started with pedometer and then used a Fitbit device. 8 participants (35%) had chosen other devices such as Fitbit, I watch, and mobile phones and 5 participants (22%) did not use any devices. The participants' adherence to the intervention was rated as "Amber" as there were some issues with recording diaries, using pedometer and performing daily exercise.

*Facilitators for participation in self-management from theme 4.* Adherence to the protocol was attributed by participants to several factors including time, knowledge and experience.

Having free time was mentioned by participants as a factor that increased their ability for self-management post-stroke.

> *"Well, I have quite a lot of time on my own, actually, so fitting things in like that is not a problem" (P6, IG).*

Previous knowledge and engagement in exercises before stroke was believed to help individual's engagement and adherence for self-management.

> *"I was quite sporty when I was younger, so the warming up and cooling down I already knew" (P4, IG).*

> *"I found them quite easy for the simple reason that before the stroke I was going to the gym for 12 years before that." (P5, IG).*

Self-motivation for improvement and independence was considered as a key for successfully engaging in self-management.

> *"You've got to be positive. You've got to get the mind right, the mindset; that's where it all starts. . . . . . because I'm motivated, it was good for me." (P1, IG).*

Social support that included family/carer support was very important to encourage participants' adherence to the programme and to ensure safety at home or in the community. Examples of family/carers support included partners' encouragement for participating in exercises sessions. In some instances, role models within the families inspired participants to engage in self-management.

> *"My family, really; they like to know how I'm doing all the time so they help me with. . . They've helped watch me with the programme" (P3, IG). "I think my wife has to take an awful lot of credit. . . . She's on the ball all the time and I'm so grateful." (P5, IG). "He's been behind me, making sure that I don't fall, things like that." (P8, CG).*

> *"So somebody coming and helping you and watching you does make a big difference." (P6, IG). "my mother-in-law, she's 88, and she said, 'I don't worry about those little pedometers,' and she walks around her block, which is a good walk, and she does that virtually every day, and she's 88, with a stick. She went, 'If I can do it, you can do it,' and it was an encouraging thing." (P7, IG).*

Contrarily, family support might negatively affect self-management if they do everything for participant.

> *"I'm lucky because I've got a strong family base and they do most of the things for me, but I don't want them to do that for me; I want to do it myself because I'm independent." (P2, IG).*

*Challenges for self-management from theme 5.* Reported challenges for the use the recording diary and pedometer included personal, technical and environmental factors such as forgetting, being busy with work, not familiar with the device, bad weather, or lost devices or documents.

*"I'm going to get better at it. I think because just generally I get so many emails and stuff, I wake up in the mornings and I'm like, 'Right, I've got to sort out work' and then I get distracted so I have to try and. . . I have to physically make the time to make sure that I put it in the diary." (P4, IG).*

*"I think the only barrier was when it was raining, you didn't want to. . ." (P9, CG).*

2 pedometers were lost and replaced during the intervention period, 1 damaged and one lost at the end of intervention.

**Attendance of the group sessions and follow up.** Overall, participants in the intervention group attended 58% of the scheduled group sessions (5 sessions for each participant over 12 weeks).

*Challenges for self-management from theme 5.* Similar to adherence, participants reported personal, technical and environmental factors that affected attendance and follow up. Some reasons that were reported in the focus groups for missing sessions were low mood, forgetting, technical issues with equipment and travelling. Some of them said they needed someone to set up the Zoom meeting for them and they had missed sessions if they did not get help.

*"I enjoyed that too (group sessions)- but just the same: I am useless with Zoom. If my husband hadn't arrived back from the GP, I still wouldn't be with you [laughs]! I am not able to use Zoom at all so he's better at it than me, I'm sure." (P3, IG).*

Another issue was that participants felt that it was hard to follow the instructor and the demonstration during the assessment or exercise sessions.

*"I think it was harder for me doing everything online just because you couldn't fully assess how I was walking." (P4, IG).*

*"I thought the group sessions were quite good except sometimes you couldn't see what the person is doing." (P5, IG).*

A 90% of the scheduled phone calls for the intervention group were successfully carried out. Participants said they were motivated for better improvement as they were encouraged in the follow up calls to do more training.

*"It gives you more motivation to move, so otherwise if we're not getting this sort of thing we won't be doing anything" (P2, IG).*

## VI) Feasibility of collecting outcome measures

Table 3 shows mean scores and standard deviations of all functional outcome measures. Overall, it was feasible to collect and retain an acceptable rate of the necessary data for all measures across the timeline of assessments. At baseline, outcome measures were collected for a 100% of participants. At 3 months and 6 months, 83.33% and 79.16% of outcome measures were collected respectively. Fig 4 shows the collection of each outcome measure across the three points of assessment.

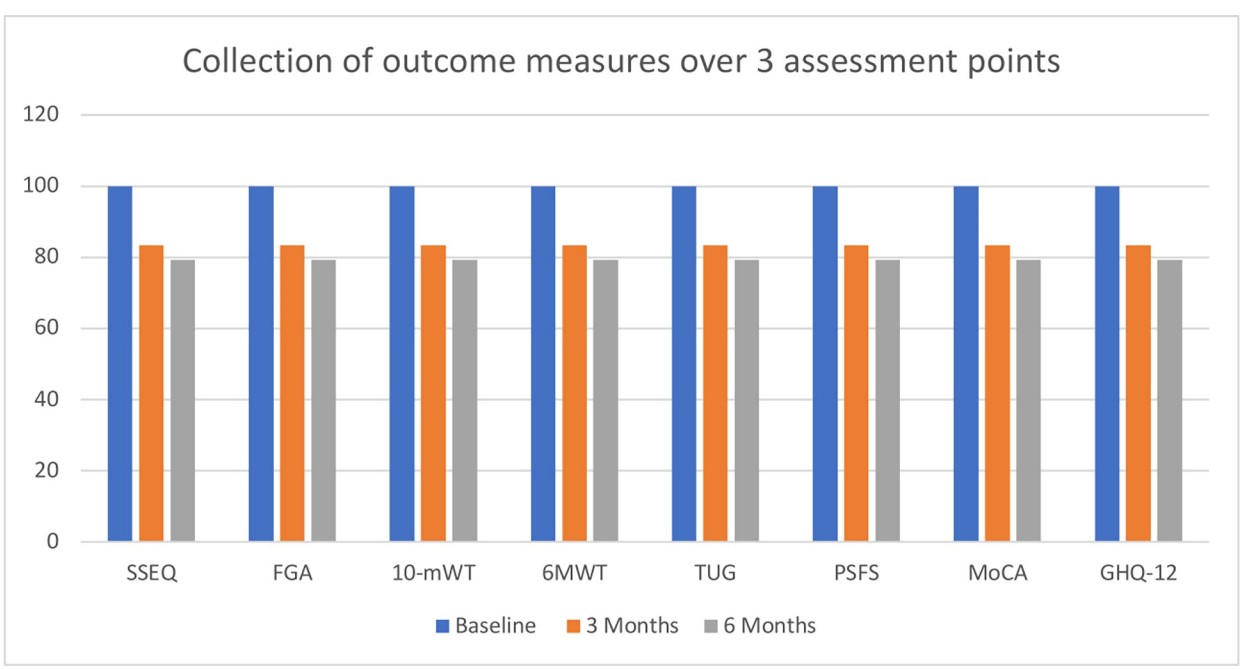

**Fig 4. Collection of outcome measures at baseline, 3 months and 6 months.**

**Participants' perspectives on selected outcome measures from theme 2.** In the focus groups, participants described their perspectives on the outcome measures used. Patients suggested that assessments were very sensible and used outcomes that are relevant to their rehabilitation needs.

*"I thought the pedometer for instance, was a relevant thing because it was giving a measurement, where in some cases you wouldn't have bothered." (P5, IG).*

*"I thought they were very, very, good actually and quite sensible, and of course you had got the comparison of the beginning and the end. So I quite agreed with that." (P6, IG).*

All assessments were completed in one session except for 2 participants who had to complete their baseline assessments over two visits because of fatigue.

## VII) Suggestions for improvement

Participants in the focus groups mentioned ideas to improve the programme in future. These suggestions included physical attendance at group meetings, addition of more group sessions and follow up calls, providing of feedback regarding the results of assessments, training for brain activities and virtual communication technology, having a physician as a part of the research team, and adapting the programme to patients with other conditions.

Overall, most of participants suggested no changes were needed to the current protocol.

*"I just think everything that's been on the courses have been helpful." (P5, IG).*

*"I think there's nothing really that you could change." (P9, CG).*

Although the study has offered physical attendance of group sessions, all participants agreed to attend them virtually. In the focus groups, some participants suggested that meetings were better when physical and more sessions (every week) can be added to the current protocol.

*"I would've liked every—once a week, me personally, because I felt a togetherness and you're meeting with other people." (P1, IG). "if occasionally all the people involved got together physically, so that they could meet and talk and show one another how they're progressing." (P7, IG).*

More communication (follow up) and providing of feedback on assessments' results were suggested.

*"Very good really, in a way. It's helped, but what I'd like is feedback from you. So, after the test, how did we do? I think the link is missing. You take information away but we don't know how we've done." (P2, IG). "I think the only thing that I think would help is more communication in one form or another." (P6, IG).*

To improve other aspects of recovery, the addition of brain activities and having a physician within the team for medical advice were suggested.

*"at the same time there is something more can be done around actually on the brain side of it, . . .. I think doing the puzzles or something like that, I think this way they can improve, yes. Just doing the puzzles also, it was activating your mind. If your mind is active then your body. That's what I believe in anyway." (P2, IG).*

*"A way I think it could be improved is maybe having a medical person advising you. . . I don't know, could we do an improvement on this?" (P2, IG).*

To improve virtual communication, a participant recommended more reminders of how to use Zoom during the course of the intervention.

*"Probably when you go around, probably show them what to do because a few of them were struggling, weren't they, to get on the Zoom at first?" (P1, IG)*

A participant suggested the adaptation of the intervention protocol for patients with other conditions.

## Discussion

This study examined the feasibility of delivering a self-management intervention to improve mobility in the community for stroke survivors after leaving hospital. The findings of the study indicate the feasibility of recruitment, retention, randomisation and blinding, implementing and collecting outcome measures for the self-management intervention targeting mobility training for the stroke survivors in the local community. Participants' perspectives show that the intervention was acceptable, applicable and they had perceived benefits in mobility outcomes. The findings also confirm the appropriateness of the study design, structure and feasibility of using randomised control trial for a future full-scale study.

It was feasible to recruit participants from multiple teams within NHS sub-acute care facilities and community services however at a lower rate due to disruptions due the pandemic sequelae. Also, as the study provided an option for online recruitment and virtual delivery of

intervention, it was feasible to recruit and successfully deliver the intervention for 2 participants from other cities within England. The study provided qualitative evidence about the effectiveness of the SM intervention for the improvement of mobility and walking parameters after stroke. Although the study was not powered to detect significant changes in performance outcomes, improvement in walking was a common theme perceived by participants in the focus groups. Other perceived improvements were reported in participants' independence, mental health, socialisation and participation. Our findings confirm the findings of previous studies about the utility of self-management for mobility rehabilitation post stroke albeit through qualitative and descriptive data [15,16]. The perceived improvement in the walking (speed, endurance, balance), self-efficacy and participation were also reported in previous studies. However, sustaining of these improvements on long-term scale need further assessment along with well-powered efficacy studies.

The recruitment plan included strategies that can help obtain the proposed sample size (90 participants) that was estimated based on a moderate sample size required for a feasibility study [46]. However, the target recruitment rate was not reached due to reasons such as impact of the pandemic on health services, fear of participants getting infected and hence reducing admissions of moderately affected patients, lack of technology to join the study virtually and other severity of issues related to stroke [14,47]. Most important reason for rejection was that a large proportion (34%) of screened patients were excluded because of mobility being too poor or too good. The participants in early stages might have been deemed ineligible due to reduced rehabilitation potential against the background of the early discharge policy (due to COVID) that was in effect at the time of screening. Similar issue of low recruitment because of clinical judgment was previously reported in a study examining the feasibility of self-management in stroke rehabilitation [20]. Authors of that study claimed that some of patients who were potentially eligible for the programme had been excluded based on care team perception of potential poor recovery or unwillingness to participation in the programme. Moreover, it has been suggested that due to the high workload of stroke rehabilitation clinicians, a therapist's role in recruitment for clinical trials can be challenged [48]. For more effective recruitment, research bodies should apply effective strategies providing time and resources. Recommendations for better recruitment include employing of research assistants, combination of multiple strategies of recruitment (i.e. involving other clinicians in the recruitment, flyers, newspapers, online advertisement), piloting and evaluation of the process to solve problems and deal with challenges [48,49].

Other reasons for exclusion represented various health problems a stroke survivor with mild impairment might have. Current literature suggested that not all patients with mild disability after stroke have a rapid and complete recovery as it had been commonly believed [50]. Mild disability post stroke has shown detrimental influence on people's participation in daily activities, psychosocial status and quality of life [51–54]. This notion was also confirmed by participants in our study who considered the consequences of stroke such as fatigue, memory and mental status as challenges for engaging in self-management. These challenges could be a reasons for participants' drop off and non-adherence to the follow up in addition to previously reported reasons (i.e. death, moving to another country, social issues, deterioration of health). From another aspect, the qualitative data suggested that engagement in self-management after stroke might require time for adjustment for some participants. This adjustment period is possibly required due to the stress and confusion that stroke survivors and their families might experience at the time of discharge from acute care. This warrants a better support for smooth transition to community where they can then start self-management [55,56]. Researchers should consider this stressful transition in addition to other functional and psychosocial

factors when communicating with participants and allow for adjustment period as a part of plan if needed.

This study excluded patients with severe impairments or co-morbidities that affected their safety and functional ability. Safety assurance is very important for the delivery of any SM type intervention especially when there is minimal supervision of formal therapists. Stroke is related to various health issues related to safety and some of them might not be commonly assessed in the early stages. For instance, stroke impact on vision and hearing are not often reported or assessed after stroke [57,58]. Also, cognitive impairments could affect 30–50% of stroke survivors to various extent. It might be worthy to consider including these people in self-management programmes with some consideration of social and environment support in future studies [59]. We had some concerns about participations of some patients (eligible from mobility wise) who referred to the study but had some other issues with cognition, vision or hearing problems. In this study although the MoCA was considered as a standardised measure of cognitive status of participants, a decision to include people with some level of cognitive problem was negotiated with their therapists. Participants with a relatively low cognitive capacity (MoCA<18) were included if they had a familiy member who can help with communication and adherence to the daily protocol. As there is no cut-off assessment for inclusion of some participants with hearing or vision impairments loss, cases were individually assessed and discussed with clinical therapists. Patients with mild to moderate impairments were included if they had a carer/family member and only those people with severe impairments that affect their safety were excluded.

Finding of this study confirms the finding of other studies about the role of family and carers in supporting self-management [16,60]. This role has been considered as essential to manage challenges at home and for reintegration in the community after discharge from hospital as it can increase the likelihood of improvement in functional and participation outcomes. For participants who joined our study remotely, it was easier and more convenient for individuals who had a carer/family member to help carrying out the online assessments and meetings. Although the support of relative/carers has been mainly considered as a facilitator for SM and reintegration in the community, in some cases too much support might limit patient's skills for SM. This idea was mentioned by some participants in the study and discussed earlier in the literature [61]. A clear description of a carer role in the SM process might be suggested to facilitate patient's independence and ensure the ideal level of social support for the successful uptake of SM interventions.

Findings of the study show a high level of acceptance and appreciation of the participants for the support they had received during the course of the intervention. Even the participants in the control group who only received an education session at the beginning of their enrolment expressed their appreciation of information provided and the provision of pedometer to track their daily steps and setting goals for rehabilitation. Previous studies show different level of patient's satisfaction with community services in the current practice. Issues such as long waiting for services, staff shortage and inadequate amount of delivered therapy were shown to affect the quality of rehabilitation post stroke [15,62]. Lack of continuity in care can increase the pressure on survivors and their families to adjust to life after discharge from hospital care and might require them to consider other options for care including private therapy. In this study, 3 participants indicated they hired private physiotherapists before joining the programme as they were waiting for community services or thought they needed more therapy than what they had received. However, it might not be feasible for every individual to have access to private care highlighting a gap in health inequality. Self-management interventions like the proposed are intended to fulfil survivors' need for continuity in care in the community with minimal resources and pressure on the NHS.

In this study, the remote delivery of intervention was suggested to suit the social distancing policy because of the pandemic, additionally the study provided the option of face-to-face delivery of the intervention. Most participants agreed to participate in the remote sessions for delivery of the education and group meetings via Zoom. In the focus groups, some participants mentioned some challenges they had experienced during their participation around remote sessions. This included poor IT skills and resources (devices and internet), less effective communication, and safety concerns. These issues have been acknowledged in rehabilitation literature and recommendations include having a facilitator (not necessarily to be a professional) who can help with service delivery, providing smart devices, or arranging for physical meetings to meet patients' preferences [14,63].

The recording diary was used in this study as an outcome measure for the recording of daily training and number of steps per day. Data collected from recording diaries and compliance of participants for their use indicated a concern about the compliance of participants in completing the recording diary. In focus groups, several reasons were mentioned for low compliance such as missing documents or forgetting or skipping the recording due to memory issues, and low mood or other stroke related physical consequences. Data collected from the recording diaries can be collected objectively using advanced devices such as pedometers or accelerometers, however, the use of recording diary was meant to emphasise the role of self-monitoring and autonomy a participant requires to achieve a previously selected goal. Similar issues with participant's compliance with recording diary have been discussed in the literature suggesting the use of electronic diaries with 'compliance-enhancing' options [64]. Further, phone reminders or email reminders can be used to improve compliance.

Education sessions were mainly delivered individually for each participant in the two groups. Just 2 sessions were delivered for a group of two participants. From our experience, it was not clear if the delivery of sessions in groups has more advantage than individual delivery. However, a recent study about the education for stroke survivors indicated that participants preferred the delivery of stroke education in groups [65]. The same study has also suggested the need to adapt content of education session for each individual based on their needs.

## Strengths and limitations

This study included stroke survivors with a wide range of clinical and demographic characteristics to examine the feasibility of the SM intervention making it applicable for a wider group of people. Lessons from previous self-management studies were considered during the development and delivery of this intervention including patient involvement in therapy planning, individualisation of plans, patient's safety, reduction of formal supervision of therapists. As the study took place during COVID-19 pandemic, the protocol of the original study was amended to tailor to the new policy in service delivery and applying some safety measures. This included the addition of remote rehabilitation, and strategies for online communication. The exercise booklet provided ability to tailor exercises within the intervention since it had three level of exercises considering that participants' ability for recovery can vary post-stroke.

The study used a mixed methods design to examine the feasibility of the intervention accounting for subjective and objective outcomes which is recommended in the examination of rehabilitation interventions [44]. Using mixed methods approach helped to account for individual factors such as functional capacity, and psychosocial factors affecting patient's participation in the study [8,9]. Moreover, in the process of goal setting and action planning for daily training, these factors were considered by the research therapist for each participant.

There were some limitations in this study. Firstly, the recruitment rate of the participants was affected by the impact of pandemic on health care system. As we did not screen patients

and only relied on teams of care, we cannot be sure of specific reasons for exclusion at screening level. It might be suggested for the definitive trial that recruitment of participants must be from a wider range of stroke care facilities to increase the possibility of having a large sample. Also, future research must investigate the residual impact of the pandemic on current policy and practice in accounting and referring of eligible patients to self-management programmes or research. Also, as the study only did focus groups with patient participants to collect patient perspectives and to discuss issues of low recruitment from NHS sites, it might be useful to include therapists who screened patients for the study in focus groups to have a better understanding of the recruitment issues they faced. Secondly, because of the nature of the intervention, it was not possible to blind the participants, however, to minimise bias, all three assessors were blinded, and effort was made to avoid communication between participants from both groups during the intervention. Although blinding has been suggested in clinical research, there has been a debate about the value of measuring the successfulness of blinding. This debate was based on the findings of some pharmaceutical studies where the interpretation of the success of blinding could problematic or misleading [66]. We sought the measuring of blinding success as an important element for the feasibility of implementing the self-management intervention in the community considering the influential impact of personal bias on study outcomes. To ensure double blinding in future studies, a cluster RCT design might be used to assign participants to a study group based on their care settings. Thirdly, there were some issues that affected participants' adherence to the intervention. In this study, some participants were allowed to use their own smart phones/watches that they were familiar with to track their activities. However, the accuracy of data collected by different devices might be considered in future studies focusing on physical activity level in particular [67]. Lastly, as the study had a relatively small sample size, it was not powered enough to detect differences in outcome between study's arms or over time, and so the patterns of means were not interpreted. A future study that is adequately powered is warranted to examine any improvements in the outcome measures.

## Conclusion

The self-management intervention seems feasible for implementation for stroke survivors in the community in England. Participants were within 6 months of discharge from hospital (mean time = 4.3 months) except one person who started the programme 7 months after discharge. There were 16 participants who lived with a carer/family member. Participants in the intervention group appreciated the support provided and desired to continue following the exercise protocol after finishing their participation. The study was not powered enough to draw a conclusion about the efficacy of the program and a future full-scale study is warranted.

## Supporting information

**S1 File. Focus group coded for Intervention and control groups.**
(PDF)

**S2 File. Protocol-Template-Non-CTIMPStudies-v10.**
(DOCX)

**S3 File. CONSORT extension for pilot and feasibility trials checklist.**
(DOC)

## Author Contributions

**Conceptualization:** Ahmad Sahely, Andrew Soundy, Sheeba Rosewilliam.

**Data curation:** Ahmad Sahely, Andrew Soundy, Sheeba Rosewilliam.

**Formal analysis:** Ahmad Sahely, Andrew Soundy, Sheeba Rosewilliam.

**Investigation:** Ahmad Sahely, Andrew Soundy, Sheeba Rosewilliam.

**Methodology:** Ahmad Sahely, Andrew Soundy, Sheeba Rosewilliam.

**Project administration:** Ahmad Sahely, Carron Sintler, Andrew Soundy, Sheeba Rosewilliam.

**Resources:** Ahmad Sahely, Carron Sintler, Andrew Soundy, Sheeba Rosewilliam.

**Software:** Ahmad Sahely, Sheeba Rosewilliam.

**Supervision:** Carron Sintler, Andrew Soundy, Sheeba Rosewilliam.

**Validation:** Ahmad Sahely, Andrew Soundy, Sheeba Rosewilliam.

**Visualization:** Ahmad Sahely, Andrew Soundy, Sheeba Rosewilliam.

**Writing – original draft:** Ahmad Sahely.

**Writing – review & editing:** Andrew Soundy, Sheeba Rosewilliam.

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
