## [Decision Letter · Decision Letter 0]

7 Sep 2023

PONE-D-23-15088Feasibility of a self-management intervention to improve mobility in the community after stroke (SIMS): a mixed-methods pilot studyPLOS ONE

Dear Dr. Sahely,

Thank you for submitting your manuscript to PLOS ONE. After careful consideration, we feel that it has merit but does not fully meet PLOS ONE’s publication criteria as it currently stands. Therefore, we invite you to submit a revised version of the manuscript that addresses the points raised during the review process.

We look forward to receiving your revised manuscript.

Kind regards,

Reindolf Anokye

Academic Editor

PLOS ONE

Journal Requirements:

Reviewers' comments:

Reviewer's Responses to Questions

**Comments to the Author**

1. Is the manuscript technically sound, and do the data support the conclusions?

Reviewer #1: Yes

Reviewer #2: Yes

2. Has the statistical analysis been performed appropriately and rigorously? 

Reviewer #1: Yes

Reviewer #2: Yes

3. Have the authors made all data underlying the findings in their manuscript fully available?

Reviewer #1: Yes

Reviewer #2: Yes

4. Is the manuscript presented in an intelligible fashion and written in standard English?

Reviewer #1: Yes

Reviewer #2: Yes

5. Review Comments to the Author

Reviewer #1: This study examined the feasibility of delivering a self-management intervention to improve mobility in the community for stroke survivors after leaving hospital.

We look forward to further studies in the future.

Reviewer #2: Stroke and its rehabilitation is an important topic to the research area

This manuscript is well written and good formulated

These comments to be considered please:

1- discussion: authors need to add a short note about causes of high rate of drop off and non adherence among patients for follow up so, the number of participants had dropped in the end of the study

Limitations: please add relative small number of patients to the limitations of the study

6. PLOS authors have the option to publish the peer review history of their article (what does this mean?). If published, this will include your full peer review and any attached files.

Reviewer #1: No

Reviewer #2: No

---

## [Author Response · Author response to Decision Letter 0]

10 Sep 2023

Dear Reviewers,

Thank you for your comments.

I hope we have responded properly to your thoughtful comments. This version of the manuscript includes responses to your comments as follow; adding a note in the discussion about the reasons for low adherence and drop off participants to the follow up, mentioning of the relative sample size in the limitations.

---

## [Editor Report · Decision Letter 1]

2 Oct 2023

PONE-D-23-15088R1Feasibility of a self-management intervention to improve mobility in the community after stroke (SIMS): a mixed-methods pilot studyPLOS ONE

Dear Dr.
Sahely,

Thank you for submitting your manuscript to PLOS ONE. After careful consideration, we feel that it has merit but does not fully meet PLOS ONE’s publication criteria as it currently stands. Therefore, we invite you to submit a revised version of the manuscript that addresses the points raised during the review process.

We look forward to receiving your revised manuscript.

Kind regards,

Reindolf Anokye

Academic Editor

PLOS ONE

Journal Requirements:

Reviewers' comments:

Stroke and its rehabilitation is an important topic to the research area

This manuscript is well written and good formulated

These comments to be considered please:

1- discussion: authors need to add a short note about causes of high rate of drop off and non adherence among patients for follow up so, the number of participants had dropped in the end of the study

Limitations: please add relative small number of patients to the limitations of the study

---

## [Author Response · Author response to Decision Letter 1]

12 Oct 2023

Dear Editor,

Thank you very much for your comment. Please note that the suggested information has been added.

Kind Regards,

Ahmad

---

## [Decision Letter · Decision Letter 2]

23 Jan 2024

PONE-D-23-15088R2Feasibility of a self-management intervention to improve mobility in the community after stroke (SIMS): a mixed-methods pilot studyPLOS ONE

Dear Dr. Sahely,

Thank you for submitting your manuscript to PLOS ONE. After careful consideration, we feel that it has merit but does not fully meet PLOS ONE’s publication criteria as it currently stands. Therefore, we invite you to submit a revised version of the manuscript that addresses the points raised during the review process.

**ACADEMIC EDITOR: **I would like to invite the authors for responding to the reviewers' comments strictly before submitting the final revised version of the manuscript 

We look forward to receiving your revised manuscript.

Kind regards,

Emad A. Aboelnasr, Ph.D

Academic Editor

PLOS ONE

Journal Requirements:

Reviewers' comments:

Reviewer's Responses to Questions

**Comments to the Author**

1. If the authors have adequately addressed your comments raised in a previous round of review and you feel that this manuscript is now acceptable for publication, you may indicate that here to bypass the “Comments to the Author” section, enter your conflict of interest statement in the “Confidential to Editor” section, and submit your "Accept" recommendation.

Reviewer #3: (No Response)

Reviewer #4: All comments have been addressed

2. Is the manuscript technically sound, and do the data support the conclusions?

Reviewer #3: No

Reviewer #4: Yes

3. Has the statistical analysis been performed appropriately and rigorously? 

Reviewer #3: No

Reviewer #4: Yes

4. Have the authors made all data underlying the findings in their manuscript fully available?

Reviewer #3: Yes

Reviewer #4: Yes

5. Is the manuscript presented in an intelligible fashion and written in standard English?

Reviewer #3: No

Reviewer #4: Yes

6. Review Comments to the Author

Reviewer #3: A two-phase sequential trial design was implemented which included a two-arm randomized controlled clinical trial. The project aimed to evaluate the feasibility of a self-management intervention to improve mobility in stroke survivors. The self-management intervention was deemed feasible.

Major revisions:

1- The abstract is not clearly written. The trial design is not clearly communicated. The abstract should mention that it was a two-phase sequential design with a randomized controlled study component.

2- Write the “Data analysis” section more clearly. Consider replacing the following statement. “The frequency counts such as means, standard deviations or ranges of values were used to summarise the analysed data.” with “Categorical data was summarized using frequencies and percentages. Continuous data was summarized using means, standard deviations or ranges.”

3- Data analysis section states, “The aim of the data analysis for the efficacy outcome measures was to evaluate the feasibility of using these outcome measures, keeping in mind that feasibility trials are underpowered to detect clinically significant treatment effects [28].” Identify and explain the statistical methods used to evaluate the feasibility of the study. Indicate if attaining a successful proportion of patients defined feasibility.

4- Patient characteristics section: Provide the standard deviation for age. In addition to the number of male participants, provide the corresponding percentage. Throughout the paragraph, in table 5.2 and throughout the document, provide percentages that correspond to the frequencies and provide standard deviations associated with means.

Minor revisions:

1- Page 17: Replace “said yes” with “agreed”.

2- Indicate if adverse events were collected according to a standardized method. In addition to frequencies, provide corresponding percentages.

3- The standard statistical term for average is mean.

4- To assist in the review process, add line numbering to the document.

Reviewer #4: (No Response)

7. PLOS authors have the option to publish the peer review history of their article (what does this mean?). If published, this will include your full peer review and any attached files.

Reviewer #3: No

Reviewer #4: No

---

## [Author Response · Author response to Decision Letter 2]

8 Mar 2024

Dear Editor and Reviewer,

Thank you for providing a productive comments.

The responses to the specific comments raised by the reviewer were reported the attached response document.

Kind Regards

Ahmad

---

## [Decision Letter · Decision Letter 3]

2 Apr 2024

PONE-D-23-15088R3Feasibility of a self-management intervention to improve mobility in the community after stroke (SIMS): a mixed-methods pilot studyPLOS ONE

Dear Dr. Sahely,

Thank you for submitting your manuscript to PLOS ONE. After careful consideration, we feel that it has merit but does not fully meet PLOS ONE’s publication criteria as it currently stands. Therefore, we invite you to submit a revised version of the manuscript that addresses the points raised during the review process.

**ACADEMIC EDITOR: **** **One of our reviewers has one more comment, so, I would like to invite the authors to respond to the raised comment before submitting the final version of the manuscript.

We look forward to receiving your revised manuscript.

Kind regards,

Emad A. Aboelnasr, Ph.D

Academic Editor

PLOS ONE

Journal Requirements:

Reviewers' comments:

Reviewer's Responses to Questions

**Comments to the Author**

1. If the authors have adequately addressed your comments raised in a previous round of review and you feel that this manuscript is now acceptable for publication, you may indicate that here to bypass the “Comments to the Author” section, enter your conflict of interest statement in the “Confidential to Editor” section, and submit your "Accept" recommendation.

Reviewer #3: All comments have been addressed

Reviewer #4: All comments have been addressed

2. Is the manuscript technically sound, and do the data support the conclusions?

Reviewer #3: (No Response)

Reviewer #4: Yes

3. Has the statistical analysis been performed appropriately and rigorously? 

Reviewer #3: (No Response)

Reviewer #4: No

4. Have the authors made all data underlying the findings in their manuscript fully available?

Reviewer #3: (No Response)

Reviewer #4: Yes

5. Is the manuscript presented in an intelligible fashion and written in standard English?

Reviewer #3: (No Response)

Reviewer #4: Yes

6. Review Comments to the Author

Reviewer #3: (No Response)

Reviewer #4: The authors demonstrated enough knowledge and understanding about the subject matter. However, adding the p and eta square values to the tables would have been greatly appreciated

7. PLOS authors have the option to publish the peer review history of their article (what does this mean?). If published, this will include your full peer review and any attached files.

Reviewer #3: No

Reviewer #4: No

---

## [Author Response · Author response to Decision Letter 3]

9 May 2024

Thank you very much for reviewing our manuscript again and for your constructive comments. We thought we should explain that we feel this comment might not be applicable to the objectives of our manuscript. 

The comment suggested that we include p and eta squared values to the tables; however, we clearly stated in the method and discussion sections that this study was not powered enough to estimate any clinical effect of the intervention as a feasibility study. Although the manuscript has demonstrated some trends toward improvement in participants’ functional and psychosocial outcomes, the results cannot offer any conclusion about the effect of the intervention.

Please note that our methodological decision was informed by the following literature that guide the development and conduct of feasibility studies.

• According to Professor Sim, between group effects from pilot studies ‘will be imprecise’ and ‘do not allow a robust decision on a main trial’. 

Sim, J. Should treatment effects be estimated in pilot and feasibility studies?. Pilot Feasibility Stud 5, 107 (2019). https://doi.org/10.1186/s40814-019-0493-7. 

• The aim of our study was to scope out design and protocol implementation as evidenced by this review by Arnold, D.M. et al. (2009) ‘The design and interpretation of pilot trials in clinical research in critical care’, Critical Care Medicine, 37(Supplement), pp. S69–S74. Available at: https://doi.org/10.1097/CCM.0b013e3181920e33.

Kindly, let’s know your thoughts on addressing this comment if you feel something still need to be done. 

Thank you again for your effort on our improving our manuscript.

---

## [Editor Report · Decision Letter 4]

26 Jun 2024

Feasibility of a self-management intervention to improve mobility in the community after stroke (SIMS): a mixed-methods pilot study

PONE-D-23-15088R4

Dear Dr. Ahmed,

We’re pleased to inform you that your manuscript has been judged scientifically suitable for publication and will be formally accepted for publication once it meets all outstanding technical requirements.

Kind regards,

Emad A. Aboelnasr, Ph.D

Academic Editor

PLOS ONE
---

## [Editor Report · Acceptance letter]

3 Jul 2024

PONE-D-23-15088R4 

PLOS ONE

Dear Dr. Sahely, 

I'm pleased to inform you that your manuscript has been deemed suitable for publication in PLOS ONE. Congratulations! Your manuscript is now being handed over to our production team.

Kind regards, 

on behalf of

Dr. Emad A. Aboelnasr 

Academic Editor

PLOS ONE